# Theoretical–Cheminformatic Study of Four Indolylphytoquinones, Prospective Anticancer Candidates

**DOI:** 10.3390/ph17121595

**Published:** 2024-11-26

**Authors:** Edgar Daniel Moyers-Montoya, María Jazmín Castañeda-Muñoz, Daniel Márquez-Olivas, René Miranda-Ruvalcaba, Carlos Alberto Martínez-Pérez, Perla E. García-Casillas, Wilber Montejo-López, María Inés Nicolás-Vázquez, René Gerardo Escobedo-González

**Affiliations:** 1Instituto de Ingeniería y Tecnología, Universidad Autónoma de Ciudad Juárez, Ave. Del Charro 450 Norte, Ciudad Juárez 32310, Chihuahua, Mexico; edgar.moyers@gmail.com (E.D.M.-M.); camartin@uacj.mx (C.A.M.-P.); 2Centro Médico de Especialidades, Av. De las Américas #201 Nte. Col. Margaritas, Ciudad Juárez 32300, Chihuahua, Mexico; al149200@alumnos.uacj.mx; 3Departamento de Mantenimiento Industrial y Nanotecnología, Universidad Tecnológica de Ciudad Juárez, Maestría en Ingeniería Industrial Sustentable, Av. Universidad Tecnológica No. 3051, Col. Lote Bravo II, Ciudad Juárez 32695, Chihuahua, Mexico; danielmarquezolivas@gmail.com; 4Departamento de Ciencias Químicas, Facultad de Estudios Superiores Cuautitlán, Universidad Nacional Autónoma de México, Avenida 1° de Mayo s/n, Colonia Santa María las Torres, Cuautitlán Izcalli 54740, Estado de México, Mexico; mirruv@yahoo.com.mx; 5Centro de Investigación en Química Aplicada, Blvd. Enrique Reyna Hermosillo No. 140, Saltillo 25294, Coahuila, Mexico; perla.garcia@ciqa.edu.mx; 6Escuela de Ciencias Químicas, Universidad Autónoma de Chiapas, Ocozocoautla de Espinosa 29140, Chiapas, Mexico; 7Escuela de Ciencias e Ingeniería, Instituto Tecnológico y de Estudios Superiores de Monterrey, Bulevar Tomás Fernández 8945, Parques Industriales, Ciudad Juárez 32470, Chihuahua, Mexico

**Keywords:** indolylquinones, in silico study, DFT, molecular docking, chemoinformatic tools

## Abstract

**Background/Objectives:** Breast cancer is a disease with a high mortality rate worldwide; consequently, urgent achievements are required to design new greener drugs, leaving natural products and their derivatives as good options. A constant antineoplastic effect has been observed when the phytoproduct contains an indole fragment. **Methods:** Therefore, the objective of this work was to carry out a thoughtful computational study to perform an appropriate evaluation of four novel molecules of the class of the 3-indolylquinones as phytodrug candidates for antineoplastic activity: thymoquinone (TQ), 2,6-dimethoxy-1,4-benzoquinone (DMQ), 2,3-dimethoxy-5-methyl-1,4-benzoquinone (DMMQ), and 2,5-dihydroxy-1,4-benzoquinone (DHQ). It is important to highlight that the obtained computational results of the target compounds were compared-correlated with the theoretical and experimental literature data previously reported of several indolylquinones: indolylperezone, indolylisoperezone, indolylmenadione, and indolylplumbagin (IE-IH, respectively). Results: The results revealed that the studied structures possibly presented antineoplastic activity, in addition to the fact that the reactivity parameters showed that two of the evaluated compounds have the option to present IC_50_ values lower than or similar to 25 mg/mL, activity like that of indolylisoperezone; moreover, they show molecular coupling to PARP-1. Finally, the prediction of the calculated physicochemical parameters coincides with the Lipinski and Veber rules, indicating that the adsorption, metabolism, and toxicity parameters are acceptable for the studied compounds, obtaining high drug score values. **Conclusions:** Finally, a comparison between the proposed molecules and others previously synthesized was appropriately performed, establishing that the synthesis of the studied compounds and the determination of their pharmacological properties in an experimental manner are of interest.

## 1. Introduction

The World Health Organization (WHO) lists cancer as the leading cause of death in the world. Regarding breast cancer [1], globally, there were (2020) more than 2.3 million new cases and 685,000 deaths from breast cancer; moreover, it is estimated that by 2040, the burden of breast cancer will increase to more than 3 million new cases and 1 million deaths each year [2]. Of particular interest, it is convenient to comment that in Mexico, it is one of the most important health challenges for both men and women; furthermore, the mortality rate has increased in recent decades [1,3] and in 2006 it became the second cause of death in women in Mexico [3,4], which was the case of the state of Chihuahua, were the mortality level is one of the highest in the country (24.8%) [5,6].

In recent years, cancer prevention via secondary metabolites has received considerable attention; therefore, phytochemistry has been welcomed as a powerful source of new drug candidates. Among the phytoproducts, and some of their derivatives produced for this purpose, it is noteworthy to acclaim the use of some quinones [7,8,9]. In this sense, it is convenient to highlight that several quinones with biological activity against cancer previously reported can be found in many vegetable specimens from the state of Chihuahua, Mexico, for example. Of particular interest in this work are thymoquinone (TQ), 2,6-dimethoxy-1,4-benzoquinone (DMQ), 2,3-dimethoxy-5-methyl-1,4-benzoquinone (DMMQ), and 2,5-dihydroxy-1,4-benzoquinone (DHQ). Additionally, these phytomolecules have presented anti-oncogenic and chemoprotective properties against various diseases [10,11,12,13,14,15,16,17,18,19,20,21].

It is also necessary to comment that many indolyl derivatives have shown interesting biological and mainly anticancer properties, highlighting the diindolylmethanes mainly used as chemoprotective and therapeutic agents against cancer, some of which are already commercially available as dietary supplements [22,23,24]. It is also notable that the indolylquinonic moiety is encountered in the class of compounds known as asteriquinones [25], with interesting biological actions, such as insulin-mimetic action, antiviral action, and the cytotoxic effect on microorganisms and cancer cells [26,27,28,29,30]; consequently, the promising uses of indolyl quinones have attracted attention for the synthesis of new molecules of this class. However, it is important to determine which molecules are of pharmacological interest to synthesize [25].

An urgent achievement is required to design new greener drugs, upholding not only efficacy but also minimizing environmental impact, in other words, an indebted effort is necessary for the development of novel pharmaceutical compounds (drugs) procuring minimization of the pollution-generating effect; however, it implicates a lengthy and labor-intensive undertaking (>10 years on average), which conveniently can be sponsored by computational methods [31].

Furthermore, the computational design establishes the properties of new drugs and their behavior on pathologies [32]. The pharmaceutical industry uses the rational design of drugs to improve the biological properties of active molecules that exist and the desired activity and also reduce undesirable secondary effects, to allow the design of new molecules with specific activities [33,34]. In this sense, the continued advances in computational chemistry allow quantum-mechanical calculation supporting the investigation in pharmacy, which contributed to raising billions in the past few years with business models that heavily rely on a combination of advanced molecular modeling. Moreover, computational chemistry benefits green chemistry in the ecology of chemical practice [35,36,37]. Pharmacological studies in recent years have made extensive use of computational methods for the development of pharmacological entities as well as for the development and testing of hypotheses [38]. The term in silico is used to describe the modeling, simulation, and visualization of both biological and medical processes using computers. The result and advances in medical informatics over the last years have achieved important progress since the use of computer technologies has favored the mapping of the human genome by currently working with SARS-CoV-2 [39]. Moreover, in pharmacology, novel interactive computer programs have supported researchers in understanding diverse techniques of pharmacology, including molecular and pharmacological parameters, ligand binding, and simulations of preparations in both humans and animals, as well as the evaluation of molecules with pharmacological potential [38].

Taking into account our research interest [27,30,40,41,42,43,44,45,46], the biological properties of the class of molecules phyto-quinones, and the interest generated for the study of indole derivatives such as indolylquinones due to their therapeutic use against various diseases, the goal of this research is to present a theoretical–computational–cheminformatic study of four indolylquinones (**IA**, **IB**, **IC**, and **ID**) derived from natural products present in plant specimens from the state of Chihuahua, Mexico; this, in order to evaluate and predict their anti-oncogenic and pharmacological potential, is proposed to appropriately inform their synthesis in the future since they have attracted the attention as potential antineoplastic agents. The corresponding structures are shown in Figure 1.

## 2. Results and Discussion

The first stage for the evaluation of the target molecules involved the optimization of the structures performing a conformational analysis and consequently, the conformer whose energy corresponded to the minimum was selected for each studied molecule. These conformers were optimized at the density functional theory (DFT) level using the B3LYP functional and the 6-311++G(d,p) basis set. The attained structures are in Figure 2.

The calculated geometrical parameters of **IA**–**1D** are displayed in Table 1. It is convenient to observe that for the target molecules, there are non-experimental X-ray partners, hence, they consider the important motif in the biological activities [27,28,30,47,48]. The experimental data both for the starter substrates and for similar indolyl derivatives in the literature [49,50,51,52] were employed to accomplish an appropriate comparison with the calculated results obtained for the target molecules.

Table 1 shows the bond length of **IA**, in comparison with the quinone ring of perezone considering the analogous structure of both; the predicted values for the target were very similar to perezone [49]. In this sense, for the indolyl moiety, the values were compared with the experimental values obtained for the X-ray diffraction of 2-ethyl-5-(3-indolyl) oxazole [50], in other words, the indolyl fragment of **IA** with the reference displayed very similar values. Regarding **IB**, **IC**, and **ID**, the quinone structures were compared with the corresponding bond length values from 2,5-dihydroxy-1,4-benzoquinone [51]; meanwhile, the indolyl moieties were compared with the X-ray diffraction of 2-ethyl-5-(3-indolyl) oxazole. The geometrical parameters obtained from **IB**, **IC**, and **ID**, quinone rings display values in agreement with the experimental data from the reference. It is convenient to note that the small difference between **IB** and **ID** can be attributed to the structural differences with the applied reference. Upon inspecting the indolyl structure, the calculated values resembled the experimental values of the reference, pointing that the DFT theory level was suitable for the studied molecules

### 2.1. Molecular Orbitals Analysis

In previous works, several indolylquinones displayed interesting cytotoxic effects against breast cancer, which were appropriately associated with their reactivity parameters, for example, the GAP energy [27,30].

In this sense, the reactivity parameters of a studied structure are acknowledged indicators for their evaluation as bioactive entities [53,54,55,56,57]; thus, Figure 3 shows the corresponding values of the energy difference between the frontier orbitals (GAP, E_LUMO_-E_HOMO_) determined for **IA**–**ID**. This parameter has been related to the reactivity of the molecule and its possible pharmacological activity [25]. These values show that the least reactive molecule corresponds to **IB**, with a ΔE_GAP_ = 63.74 kcal/mol, in other words, the most stable.

In addition, the GAP energies implied that the most reactive molecule was **IC**, ΔE_GAP_ = 49.95 kcal/mol, followed by **IA**, ΔE_GAP_ = 58.85 kcal/mol. Considering that in previous works both the GAP and the IC50 of structurally similar molecules were reported, it was considered appropriate to compare the GAP of the study molecules and those previously reported [27,30] in order to estimate the possible activity of the study molecules (**IA**–**ID**). In this way, **IA** exhibited a similar GAP value to indolylisoperezone (58.85 and 58.33 kcal/mol, respectively), and the reactivity of **IC** is bigger than the indolylisoperezone; meanwhile, **IA** is minor compared to this compound. Therefore, it can be expected that **IC** should display bigger cytotoxic activity against breast cancer than indolylisoperezone (IC_50_ = 25 μg/mL) and **IA** with similar cytotoxicity values. In the same sense, **ID** and **IB** had similar GAP energies value than the indolylplumbagin (ΔE_GAP_ = 61.63) and indolyl menadione (ΔE_GAP_ = 62.82 μg/mL), respectively. In the same way, the cytotoxicity of **ID** should be comparable with indolylplumbagin (IC_50_ = 32.28 μg/mL), and the cytotoxicity of **IB** should be like indolylmenadione (IC_50_ = 41.58 μg/mL).

To improve the reported relationship between reactivity parameters and cytotoxicity against breast cancer cells (IC_50_), as well as to expand the comparison of the target molecules with reported similar structures, additional reactivity parameters were calculated for **IA**–**ID** and indolylperezone (**IE**), indolylisoperezone (**IF**), indolylmenadione (**IG**), and indolylplumbagin (**IH**), using previously reported geometries [27]; the corresponding results are shown in Table 2 and Table 3.

Table 2 exhibits the reactivity parameters of **IA**–**ID**; in this sense, **IC** displayed both the biggest electronic affinity and hardness, and these results can be attributed to the electron-donating capability of the hydroxyl groups in the quinone ring. Meanwhile, **IB** exposed bigger chemical potential and electrophilicity values; these results reveal the best capacity to accept electrons due to the methoxy group in the quinone ring. Finally, **IA** displays big ionization energy, in agreement with the structure of the compound, which shows the absence of a strong electron-donating group to stabilize the structure.

In Table 3, the reactivity parameters calculated for **IE**–**IH** are summarized; as can be seen, the indolylplumbagin exhibits the bigger value of electronic affinity (*EA*), ionization energy (*I*), and hardness (*η*). These facts can be attributed to the presence of the aromatic system and the electronic density provided by the hydroxyl group. Additionally, indolylmenadione displays the highest values in chemical potential and electrophilicity. In contrast, the menadione shows the lowest electronic affinity, which can be attributed to the aromatic ring; meanwhile, the indolylperezone shows the lowest values in the remaining properties. It is convenient to highlight that the perezone quinone ring has a small reduction potential, which could provoke easy reductions and oxidation reactions.

Since **IE**–**IH** depicted cytotoxic data against breast cancer (*vide supra*), their reactivity parameters were correlated with the IC_50_ values to identify any trends and try to obtain an approximation to the IC_50_ of **IA**–**ID**. In this sense, Figure 4 shows the chemical reactive parameters correlated with IC_50_ and its trend line with the corresponding regression equation and the determination coefficient. The data suggest a polynomial relationship between IC_50_ and all of the calculated parameters from the previously reported indolylquinones [27].

These models do not allow the calculation of the IC_50_ of the previously reported compounds from the reactivity parameters. However, regarding the chemical potential and electrophilicity, its parameters exhibit a behavior closer to a linear relationship with IC_50_, as previously reported [58,59,60,61]. Focusing on the linear segment of the graph, an inverse proportional pattern can be observed between the chemical potential and IC_50_. However, in a biological sense, the compounds with lower chemical potential exhibit lower cytotoxicity activity. Additionally, the linear segment in the graph relating to IC_50_ and electrophilicity shows a direct proportional relationship (Figure 5). In this context, a lower value of electrophilicity corresponds to a lower IC_50_ value and, consequently, greater bioactivity.

The linear segment of the chemical potential (*μ*) and electrophilicity (*Ω*) with the IC_50_ of **IG**, **IH**, and **IF** were analyzed by linear regression. In this sense, IE data were omitted from this analysis due to their deviation from linearity. The obtained equations were used to estimate the IC50 of **IA**–**ID** from their electrophilicity and chemical potential values; their estimated IC_50_ was summarized in Table 4. The results of this analysis showed that the IC compound exhibits the lowest cytotoxicity in comparison with indolylisoperezone. Additionally, IA and ID showed similar values of cytotoxicity compared to indolylperezone and indolyl menadione, respectively.

In addition to the calculation of the GAP and the reactivity parameters, the analysis of the electron density surface was performed. This analysis allowed us to recognize the regions where the HOMO (highest occupied molecular orbital) is located, commonly associated with the capacity for electron donation, while the LUMO (lowest unoccupied molecular orbital) indicates regions of the molecule with the ability to accept electrons [62]; the contours of the molecular orbitals are depicted in Figure 6 and Figure 7. In general, the HOMO orbitals were situated both in the region of the indole fragment and partially in the conjugated double bonds of the quinone structure; these regions with π-electrons are available, allowing these substrates to act as nucleophiles.

Regarding the LUMO orbitals, these orbitals were located around the quinone ring, a situation attributed to both the effects of electron density subtraction, which are both inductive, as well as resonance by the conjugated carbonyl group. This region allows the substrates to act as an electrophile; it is convenient to highlight that this statement is consistent with studies on similar molecules [27].

Electrostatic potential surfaces serve as an indicator that enables the prediction of molecular reactivity and interactions of bioactive molecules in biochemical phenomena [63]. Some studies have reflected that the reactive regions identified in a target molecule can correspond to those involved in an electron donor–acceptor interaction phenomenon [64].

In this sense, the electrostatic potential maps of **IA**–**ID** were calculated at the B3LYP/6-311++G(d,p) level of theory; they are highlighted by a color range of +6.6 × 10^−2^ deep blue zone and −6.6 × 10^−2^ deep red zone, as shown in Figure 8.

The maps show the nucleophilic sites in red and electrophilic sites in blue; the potential values are displayed at the sites with the highest electron density. It also can be observed that the distributions present a higher electron density in the oxygen atom of the carbonyl groups within the quinone ring, where the range lies from −0.0162661 to −0.0515837; in this case, the oxygen atom in the carbonyl groups with a hydrogen bond is followed to a lesser extent in the hydroxyl and methoxy groups, whose values are between −0.0235 and −0.0482. According to other studies, these phenomena can be attributed to the presence of intramolecular hydrogen bonds [30,65]. In the case of the sites with the highest electron deficiency, it can be observed that the potentials are between +0.0626 and +0.0656, corresponding to the hydrogen attached to the nitrogen of the indole. It is important to mention that these regions have a high probability of contributing to non-covalent interactions with some amino acid residues in biological receptors.

The PASS chemoinformatics obtained data allowed us to expect different types of pharmacological activity for the target molecules; the most relevant results are shown in Table 5.

The program is based on inactive (PI) and active (PA) categories, with a probability of 95%. The analysis establishes that if the Pa value is greater than 0.7, the molecule has the probability of presenting biological activity. Contrarywise, a Pa value between 0.5 and 0.7 implies that the compound is relatively indefinite; therefore, there is a reduced probability of retrieving a pharmaceutical agent. Moreover, a Pa less than 0.5 implies that the molecule has no similarity with other pharmaceutical agents; although, it does not mean that the compound could not have a pharmacological activity. Among the highlighted activities, it is convenient to underline their potential use as agents for neurological disorders. It is also important to note that despite the results indicating antineoplasic and apoptotic activities with a Pa value less than 0.5, several experimental studies show that analogous molecules have this activity [65,66,67].

The physicochemical properties and the corresponding parameters related to Lipinski’s rules were carried out using the molinspiration chemoinformatics program; these results are summarized in Table 6. In this sense, the physicochemical properties of the studied compounds were predicted, including clog *P*, the logarithm of the partition coefficient between n-octanol and water, a property that describes the molecular hydrophobicity [68,69]. The values obtained varied from 0.38 to 2.93, and IB and IC showed the lowest values. In this sense, the studied molecules have values less than 5, suggesting a reasonable possibility to be well absorbed [70,71].

The log *S* information stands for drug solubility, which is an important property to describe the absorption process; therefore, poor solubility leads to poor absorption and bioavailability. It is important to note that commercial drugs in general exhibit log *S* values greater than −4, concerning the target molecules, and all of them exhibited values over −4, with **IC** leading to the highest one; in other words, **IC** must exhibit the best absorption and distribution in the bloodstream and the best elimination through the urinary tract [65,68,69,71]. In this sense, the lowest solubility corresponded to **IA** and consequently the lower absorption.

Lipinski’s rule of five establishes that the molecular weight of pharmacologically active compounds must ideally stay between 160 and 500 g/mol; in this sense, all the studied molecules exhibited values in agreement. Furthermore, the indolylquinones showed hydrogen bond donors ≤ 10 and hydrogen bond acceptors ≤ 5, all of them within the acceptable values defined by Lipinski’s rules [72].

Another good descriptor for absorption is the total polar surface area (TPSA), including intestinal absorption, bioavailability, Caco-2 permeability, and blood–brain barrier penetration [73]. In this regard, **IC** exhibited the highest TPSA value (90.39), while **IA** displayed the lowest 49.93, bringing into line lower solubility and molecular hydrophobicity.

Finally, the drug score (DS), the combination of drug-likeness, clog *P*, log *S*, molecular weight, and toxicity risks, is summarized into a single value that can be used to assess a compound’s potential as a drug [74]. In this sense, **IA**, **IC**, and **ID** showed higher values. The lower drug score predicted for **IB** was attributed to the high probability of mutagenic effects.

### 2.2. Docking Study of the Apoptosis Pathways

Several indolylquinones with structures analogous to the studied molecules have shown anticancer activity, like as human MDA-MB-231 breast cancer and MCF-7 breast cancer cells, which shows evidence of apoptotic pathways. However, the PASS analysis of **IA**–**ID** points out a low probability of antineoplastic and apoptotic activity. Therefore, to appropriately evaluate this potential, a molecular docking study was performed to assess the possible interaction of the target molecules with the PARP-1 protein; Table 7.

The interaction models among the studied indolylquinones and PARP-1 are displayed in Figure 9 and Figure 10.

As can be seen, Figure 9 displays the docking of the **FRM** (2-{3-[4-(4-fluorophenyl)-3,6-dihydro-1(2*H*)-pyridinyl] propyl}-8-methyl-4(3*H*)-quinazolinone) and the four indolylquinones evaluated with PARP-1. These results reveal that all molecules are docked in the active site with a similar orientation. Figure 10 depicts the interactions between **FRM** and the indolylquinones **1A**–**1D** with the amino acid residues in the active site of PARP-1. In the case of **IA**, this molecule shows a strong hydrogen bond with the tyrosine 235 residue (1.95 Å, 160.57°) between the quinone carbonyl group as acceptor and the N-H bond of the tyrosine residue (donor). Additionally, the carbonyl group acceptor ensured a high negative value for the electrostatic molecular potential (−0.0442685).

Concerning **IB**, this molecule developed two weak hydrogen bonds [75]. The first one is an N-H bond between the indolyl moiety (donor) and glycine 202 (bond length of 2.148 Å and bond angle 154.29°). Particularly, the N-H bond of indolyl moiety has a strong positive molecular electrostatic potential (+0.0633308); the second hydrogen bond corresponds to the oxygen of the methoxy group and the methionine 229 (bond length 2.191 Å and bond angle of 147.7496). The methoxy group displays a high negative value of molecular electrostatic potential (−0.0411204). A similar interaction was observed for **ID** with the same amino acid residues (Gly202 and Met229) and the same donor and acceptor groups. The donor and acceptor groups align with their molecular electrostatic potential values.

Finally, **IC** interacts with the active site involving three weaker hydrogen bonds [75], two of them with a hydroxyl group, one hydroxyl group as a donor (molecular electrostatic potential value = +0.0361708) with the aspartic acid 105 (bond length of 2.213 Å and bond angle 133.158°), and the second hydroxyl group act as a donor (molecular electrostatic potential value = −0.0205295) with isoleucine 234 (bond length of 2.237 Å and bond angle 123.509°). The third hydrogen bond is formed between the N-H of the indolyl group (molecular electrostatic potential value = +0.0450351) and the aspartic acid 109 (bond length of 2.064 Å and bond angle 153.495°). The relationship between these results suggests that the interaction models are feasible, indicating a possible anticarcinogenic effect. Additionally, three of the indolylquinones exhibited drug scores above 0.7.

Pharmacokinetic parameters are crucial for understanding the mechanisms of action of a drug, including, in this case, the indolylquinones. In this sense, Table 8 provides a detailed pharmacokinetic analysis of indolylquinones that includes parameters such as human intestinal absorption, permeability in Caco-2 cells, glycoprotein-P (a drug resistance-related protein), as well as biotransformation of drugs where the cytochrome P450 family is of vital importance, measuring its activity as an inhibitor or substrate. On the other hand, its toxicity was predicted by the AMES test, specifically, its potential carcinogenic.

The obtained results are also indicative of a high probability of intestinal absorption for these compounds, with values above 0.9900. Regarding human oral bioavailability, it shows average values, where the lowest is **IA** at 0.6571 and **ID** with the highest value at 0.8286; this indicates that the compound **IA** is less polar due to the absence of a hydroxyl group.

An important property for drug discovery is the permeability of CACO-2, which shows a high value. The lowest is for the indolyl-2,5-dihydroxy-1,4-benzoquinone with 0.7145 and the highest value is 0.8133 for **IA**, displaying the permeability of the drugs in the culture of cells in a monolayer of human colon adenocarcinoma, which is used for drug discovery [76].

In this sense, the prediction of the target molecules shows permeability in cases of **IA**, **IB**, and **ID** with different probability values; however, **IC** displays negative results to Caco-2 permeability. 

Regarding distribution, P-glycoproteins are key transporters found in various tissues; one of its functions is clear xenotoxins against concentration gradients playing an important role in transporting small molecules in vital areas. P-glycoprotein is also overexpressed in many multidrug-resistant cancer cells, and its inhibition can help overcome drug resistance [77,78]. Indolyl-2,3-dimethoxy-5-methyl-1,4-benzoquinone is expected to be a good substrate for these proteins. Additionally, all the explored compounds exhibited a high probability of penetrating the blood–brain barrier, a protective structure against foreign substances, indicating that the four compounds are good for distribution [79].

The metabolism parameters, Table 9, showed the behavior of indolylquinone as a substrate or inhibitor in the target molecules, in the most important isoform of cytochrome P450, which is involved in the metabolism of therapeutic drugs. 

During the biotransformation process, drugs break down the molecules into more soluble ones, since they play an important role in the therapeutic and pharmacokinetic action of the drug. The evaluation of the molecules as substrates suggests that all of them maintain the 3A4 isoform with medium probabilities, and they also act as inhibitors of the 1A2 and 2C9 isoforms with medium probabilities [30]. In the toxicological parameter, Table 9, it can be seen that indolyl-2,6-dimethoxy-1,4-benzoquinone is toxic in the AMES test, implying that this compound can be mutagenic with the medium probability [80]. However, the evaluated compounds are expected to a low probability of carcinogenicity or ocular corrosion.

## 3. Materials and Methods

### 3.1. Optimization of the Structures and GAP Energy

The studied molecules were constructed with standard bond lengths and bond angles using the PC program Spartan06 [81]. Then, a conformational analysis was performed and the most stable conformations were selected. The minimum energy conformers were fully optimized using the DFT level of theory in the Gaussian 09 [82] program, using the Becke three-parameter hybrid functional (B3LYP) [83,84], also employing the 6-311++G(d,p) basis se [85,86,87,88,89,90] that includes the split and diffuse valence functions. The highest occupied molecular orbital and lowest unoccupied molecular orbital (HOMO−LUMO) gap describe the dynamic stability of molecules (Equation (1)); these values of the orbital energy were acquired using the same level of theory.
(1)$$E_{gap}=E_{Homo}-E_{Lumo}$$

### 3.2. Reactivity Descriptors

The reactivity descriptors chemical potential (*µ*), hardness (*η*), and electrophilicity (*ω*) were achieved, applying their respective equation [61,62,63,64,65] as follows:(2)$$\mu =-\frac{\left(I+EA\right)}{2}$$
(3)$$\eta =\frac{I-EA}{2}$$
(4)$$\omega =\frac{\mu^{2}}{2n}$$
where *EA* is the electron affinity and I represents the ionization energy; *I* and *EA* were obtained from the neutral, cationic, and anionic forms using its electronic energies at the same theory level.

### 3.3. Molecular Electrostatic Surface

The molecular electrostatic potential map (MEP) is a robust predictive and interpretative tool for chemical and intermolecular interactions. Also, MEP is commonly employed to carry out a reactivity map showing the most probable molecular regions to realise an electrophilic attack by a reagent to the substrate [91]; it provides information to understand the shape, size, charge density, delocalization, and site of chemical reactivity of the molecules. MEP maps can be achieved by mapping electrostatic potential onto the total electron density with a color code [62,92]. In this sense, a MEP contour map provides a simple way to predict how different geometries could interact [65]. This property was determined using the DFT (B3LYP/6-311++G(d,p) method.

### 3.4. Biological Activity Predictions

The biological activity predictions were carried out using the chemoinformatic PASS server, which predicts over 3500 kinds of biological activities. The predicted biological activity profile was obtained from the structural formula of compounds and was based on the analysis of structure–activity relationships for more than 250,000 biologically active substances, including drugs, drug candidates, primes, and toxic compounds [33,34,35,68,69,70].

### 3.5. Toxicological and Physicochemical Properties Prediction

The physicochemical properties for target molecules display relevant information regarding the facility of a drug to interact with the aminoacid residues inside cells or the membrane receptors. log *P* and log *S* were obtained by adding the contributions of every atom based on its properties. The drug likeness approach is based on a list of about 5300 distinct substructure fragments with associated drug-likeness scores. The drug likeness is calculated by employing the score values of those fragments present in the molecule under investigation. The toxicological risk prediction process relies on a precompiled set of structural fragments that give rise to toxicity alerts in case they are encountered in the structure currently drawn. The toxicological risk and physicochemical properties of the studied molecules were obtained using the OSIRIS property explorer [38,71,72].

### 3.6. Docking Study in the Apoptosis Pathways

Molecular docking predicts the binding capability of a molecule to an examined protein by developing a relatively stable complex. The expected ligand orientations allow the prediction of the preferred binding conformations and affinity between the proteins and small molecules as ligands [93,94]. The molecules with high-binding affinities are recognized as potential candidates for experimental validation. Hence, in this work, the molecular docking simulations were performed using the three-dimensional crystal structures of the protein PARP-1 (1UKO) obtained from the RCSB, Protein Data Bank [95,96]. Regarding the ligand structure, the optimized structure previously obtained for the indolyl quinones studied was used. The software Autodock 4.2 was employed to carry out the docking studies [97,98], which has been extensively used for its good correlations among docking results and experimental values [99]. The ligands were not rigid and allowed torsion in the first blind docking processes. The grid box was 70 Å × 70 Å × 70 Å, encompassing the ligand binding cavity of the protein with points separated by 0.375 Å. All docking simulations were conducted using the hybrid Lamarckian genetic algorithm with a maximum of 10^7^ energy evaluations and a population size of 150. All of the other parameters were maintained at their default settings. The docking results were ranked according to the binding free energy to compare with the leader drug. The binding modes with the more negative binding free energy and maximum hydrogen bonding number were selected as the optimum docking conformation.

### 3.7. Pharmacological and Metabolic Properties

The absorption and metabolic properties of the studied compounds were calculated by using the admetSAR server [100,101], which predicts about 50 ADMET endpoints using a chemo-informatics-based toolbox, called the ADMET-Simulator, which integrates high-quality and predictive QSAR models.

## 4. Conclusions

An insightful in-silico study to evaluate the pharmacological and therapeutical potential activity for a set of four indolylquinones designed from natural quinones was appropriately accomplished. It is important to note that the obtained results suggest that **IC** and **IA** exhibited both the best reactivity in addition to the IC_50_ with higher cytotoxicity. The calculated drug score was bigger than 0.7–0.9. Moreover, the predicted bioactivity of the studied molecules **IA**–**ID** displays acute neurologic disorders treatment.

Finally, the docking studies are indicative that **IA** or **IC** agree with a good energy interaction with the active site of PARP-1. Because breast cancer is an important disease in Chihuahua, Mexico, and the world, indolylquinones attract attention as potential antineoplastic agents, so the design and corresponding evaluation of new molecules of this class will allow us to establish which of them may present greater activity. Consequently, the comparison between the proposed structures and others previously synthesized was appropriate, determining that given their potential activity, the synthesis of these new indolylquinones using the green chemistry protocol and the experimental study of their pharmacological properties is of interest as alternatives for the treatment of breast cancer.

## Figures and Tables

**Figure 1 pharmaceuticals-17-01595-f001:**
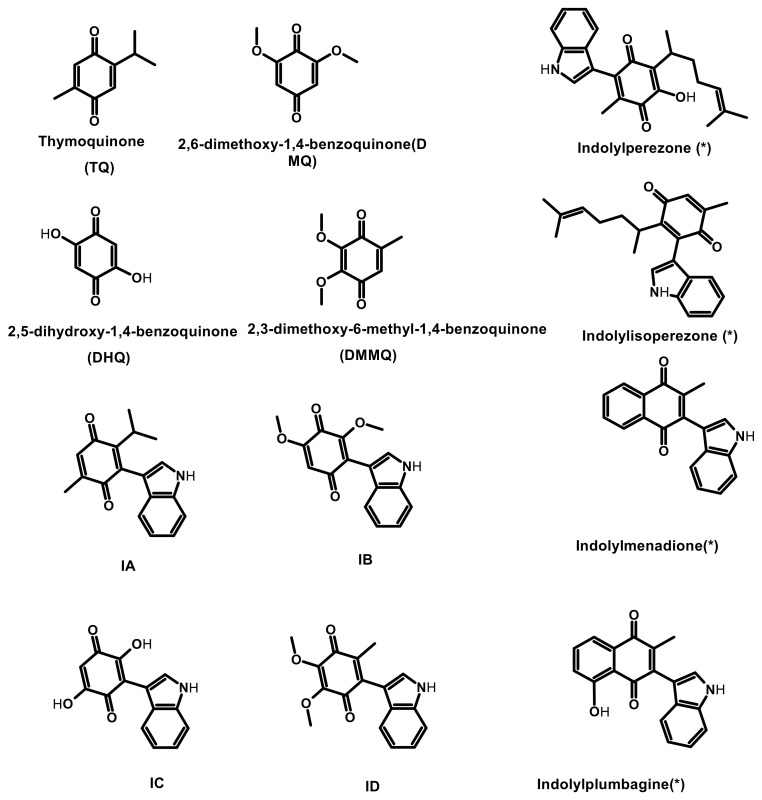
Structures of natural quinones used as a basis for the proposed indolylquinones (TQ, DMQ, DHQ, DMMQ), the indolylquinones studied (IA–ID), and the reference molecules (*): indolylperezone, indolylisoperezone, indolylmenadione, and indolylplumbagin (IE–IH, respectively).

**Figure 2 pharmaceuticals-17-01595-f002:**
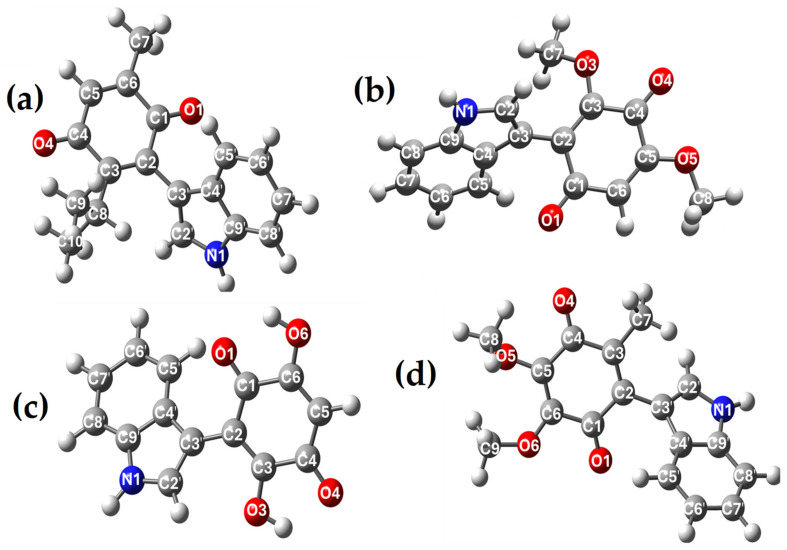
Structures optimized by B3LYP/6-311++G(d,p) of the indolyl derivatives of (**a**) thymoquinone (IA), (**b**) 2,6-dimethoxy-1,4-benzoquinone (IB) (**c**) 2,5-dihydroxy-1,4-benzoquinone (IC), and (**d**) 2,3-dimethoxy-5-methyl-1,4-benzoquinone (ID).

**Figure 3 pharmaceuticals-17-01595-f003:**
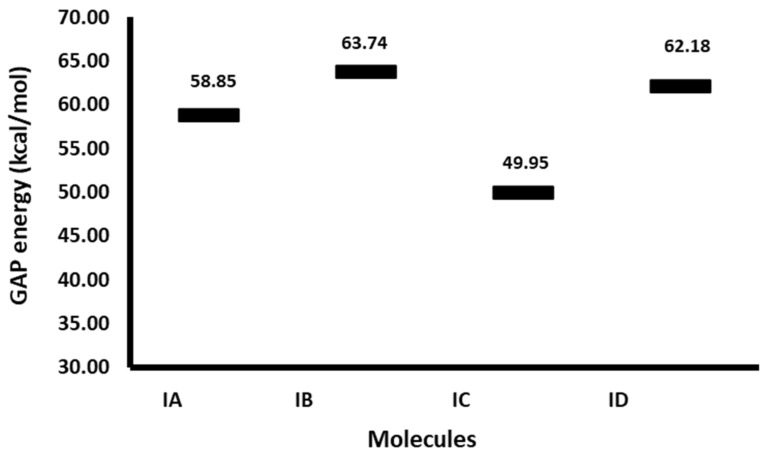
GAP energy plot (kcal/mol) for the target molecules.

**Figure 4 pharmaceuticals-17-01595-f004:**
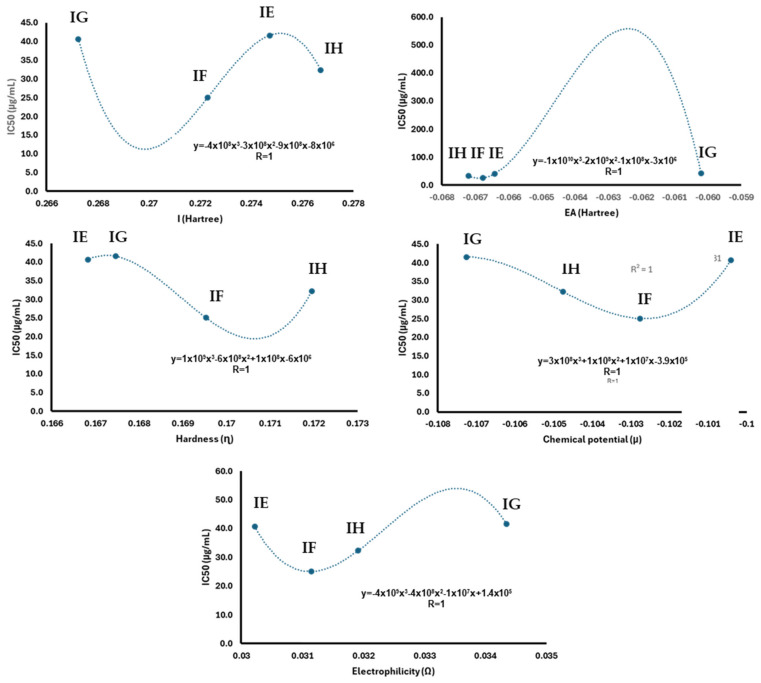
Regression analysis from reactivity parameters with IC_50_ from indolylquinones previously reported [27].

**Figure 5 pharmaceuticals-17-01595-f005:**
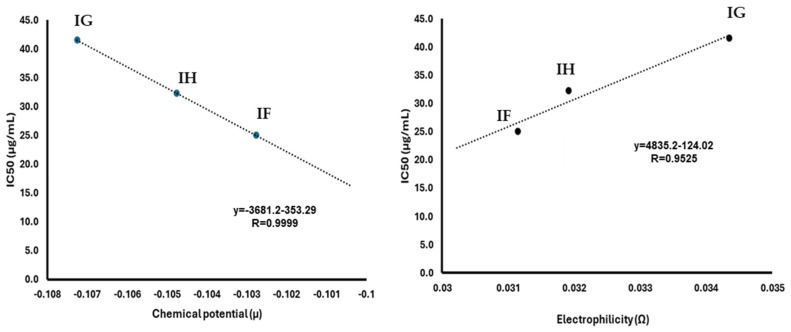
Regression analysis from chemical potential and electrophilicity parameters with IC_50_ from indolylquinones was previously reported [27].

**Figure 6 pharmaceuticals-17-01595-f006:**
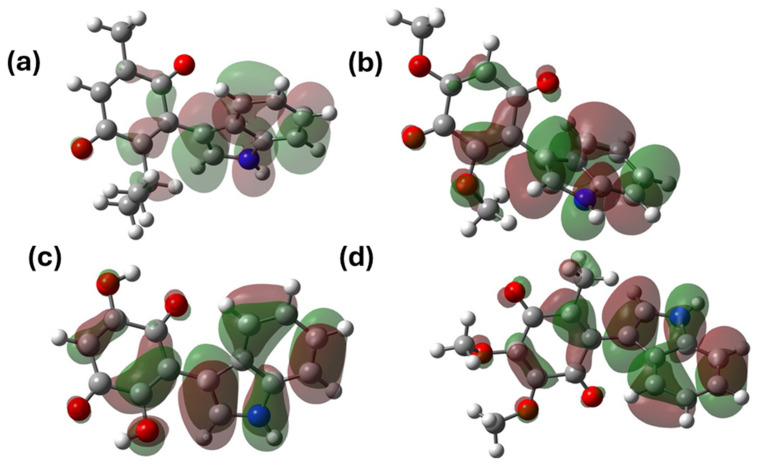
HOMO molecular orbitals of the indolic derivatives of (**a**) thymoquinone, (**b**) 2,6-dimethoxy-1,4-benzoquinone, (**c**) 2,5-dihydroxy-1,4 -benzoquinone, and (**d**) 2,3-dimethoxy-5-methyl-1,4-benzoquinone.

**Figure 7 pharmaceuticals-17-01595-f007:**
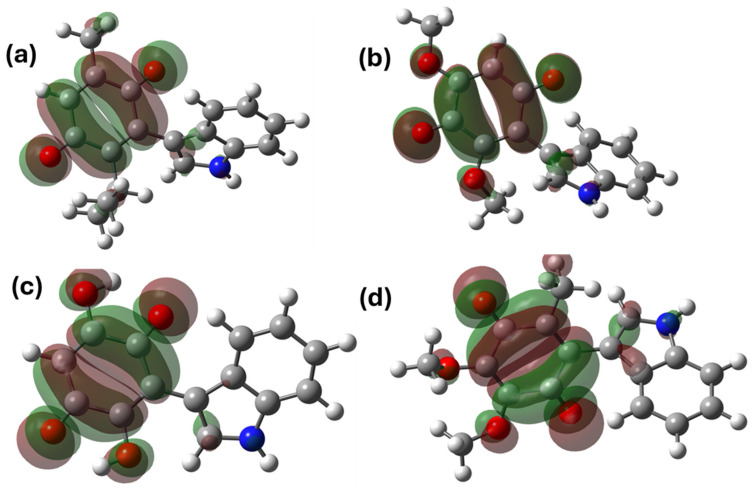
LUMO orbitals of the indolyl derivatives of (**a**) thymoquinone, (**b**) 2,6-dimethoxy-1,4-benzoquinone, (**c**) 2,5-dihydroxy-1,4 -benzoquinone, and (**d**) 2,3-dimethoxy-5-methyl-1,4-benzoquinone.

**Figure 8 pharmaceuticals-17-01595-f008:**
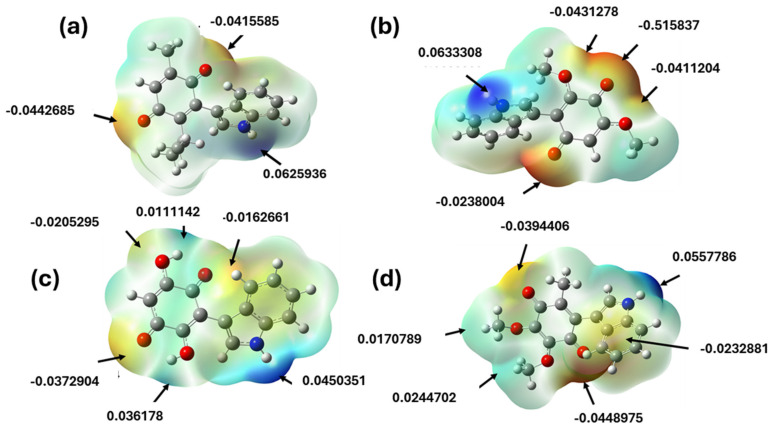
Molecular electrostatic potential maps of the indolylderivatives of (**a**) thymoquinone, (**b**) 2,6-dimethoxy-1,4-benzoquinone, (**c**) 2,5-dihydroxy-1,4-benzoquinone, and (**d**) 2,3-dimethoxy-5-methyl-1,4-benzoquinone.

**Figure 9 pharmaceuticals-17-01595-f009:**
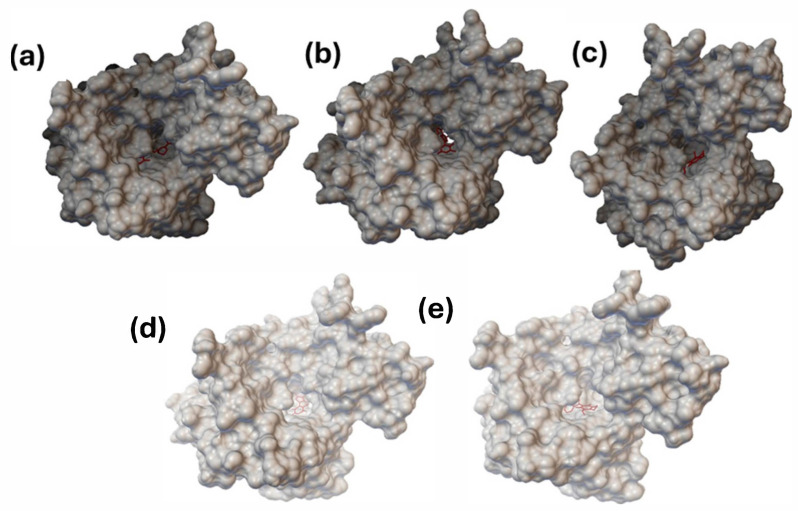
PARP-1 interaction with (**a**) **FRM** (**b**) **IA** (**c**) **IB** (**d**) **IC** (**e**) **ID**.

**Figure 10 pharmaceuticals-17-01595-f010:**
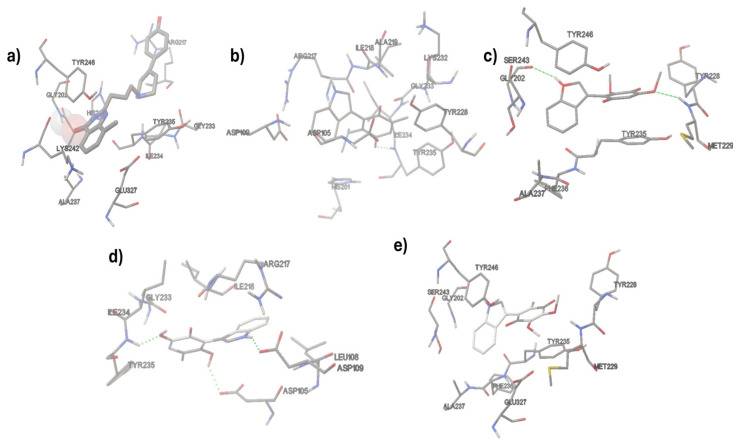
Amino acid residue interaction of the active site of PARP-1 with (**a**) FRM (**b**) IA (**c**) IB (**d**) IC (**e**) ID.

**Table 1 pharmaceuticals-17-01595-t001:** Theoretically and experimentally selected bond length (in Angstroms) for the target molecules.

Bond	Theoretical Bond Length (Å)	Experimental Bond Length (Å)	Bond	Theoretical Bond Length (Å)			Experimental Bond Length (Å)
	IA			IB	IC	ID	
C1-O1	1.219	1.226 ^a^	C1-O1	1.224	1.214	1.213	1.232 ^c^
C1-C2	1.505	1.469 ^a^	C1-C2	1.497	1.481	1.501	1.449 ^c^
C2-C3	1.361	1.334 ^a^	C2-C3	1.362	1.363	1.355	1.332 ^d^
C3-C4	1.499	1.498 ^a^	C3-C4	1.508	1.495	1.497	1.494 ^d^
C4-C5	1.477	1.469 ^a^	C4-C5	1.504	1.448	1.481	1.485 ^d^
C4-O4	1.224	1.22 ^a^	C4-O4	1.209	1.231	1.255	1.208 ^d^
C5-C6	1.340	1.332 ^a^	C5-C6	1.471	1.348	1.36	1.352 ^c^
C6-C7	1.498	1.507 ^a^	C3-O3	1.344	1.342		
C3-C8	1.520	1.499 ^a^	C5-O5	1.337		1.365	
C8-C9	1.542		C3-C7			1.502	1.502 ^d^
C8-C10	1.544		C6-O6		1.338	1.336	1.328 ^b^
C2-C3′	1.476	1.448 ^b^	C2-C3′	1.473	1.464	1.471	1.448 ^b^
C2′-C3′	1.374	1.375 ^b^	C2′-C3′	1.374	1.38	1.376	1.375 ^b^
C2′-N	1.377	1.363 ^b^	C2′-N	1.378	1.369	1.375	1.363 ^b^
C9′-N	1.382	1.370 ^b^	C9′-N	1.382	1.383	1.383	1.370 ^b^

^a^ Obtained from Soriano-García et al., 1986 [49]; ^b^ obtained from Noltemeyer et al., 1982 [50]; ^c^ obtained from Cowan et al., 2001 [51]; ^d^ Obtained from Esser et al., 2002 [52].

**Table 2 pharmaceuticals-17-01595-t002:** Reactivity parameters for **IA**–**ID**.

Parameters		IA	IB	IC	ID
Energy (Hartrees)	Neutral	−901.580391	−973.386146	−894.79862	−1012.70919
	Positive	−901.303741	−973.111114	−894.526526	−1012.4383
	Negative	−901.642824	−973.439368	−894.86798	−1012.76554
Reactivity parameters	*EA*	−0.06243303	−0.05322233	−0.06936025	−0.05634553
	*I*	0.27665028	0.27503243	0.27209414	0.27089329
	*η*	0.16954166	0.16412738	0.17072719	0.16361941
	*μ*	−0.10710862	−0.11090505	−0.10136695	−0.10727388
	*Ω*	0.03383315	0.03747068	0.03009262	0.03516602

**Table 3 pharmaceuticals-17-01595-t003:** Reactivity parameters from indolylperezone (**IE**), indolylisoperezone (**IF**), indolylmenadione (**IG**), and indolylplumbagin (**IH**) and its IC_50_ dose.

Parameters		IE	IF	IG	IH
Energy (Hartrees)	Neutral	−1172.235961	−1172.227355	−937.3004684	−1012.561843
	Positive	−1171.968715	−1171.955049	−937.0257474	−1012.285115
	Negative	−1172.302376	−1172.294128	−937.3606764	−1012.629042
Reactivity parameters	*EA*	−0.06641433	−0.06677303	−0.060208	−0.067199
	*I*	0.26724651	0.27230539	0.27472097	0.2767278
	*ηH*	0.16683042	0.16953921	0.167464485	0.1719634
	*μΜ*	−0.10041609	−0.10276618	−0.107256485	−0.1047644
	*Ω*	0.030220481	0.031145856	0.034347443	0.031912545
Dose	µg/mL	40.64	25.06	41.58	32.28

**Table 4 pharmaceuticals-17-01595-t004:** Predicted IC_50_ in μg/mL for **IA**–**ID**.

Reactivity Properties	Equation	IA Predicted IC_50_	IB Predicted IC_50_	IC Predicted IC_50_	ID Predicted IC_50_
Chemical potential	IC_50_ = −3681.2 µ − 353.29	40.998	54.974	19.862	41.607
Electrophilicity	IC_50_ = 4835.2 Ω − 124.02	39.570	57.1582	21.484	46.015

**Table 5 pharmaceuticals-17-01595-t005:** Some relevant biological activities of the target compounds acquired by PASS.

Activity/Molecule	Probability of Activity (Pa)			
	IA	IB	IC	ID
Apoptosis agonist	0.404	0.299	0.445	0.412
Antineoplastic	0.202	0.429	0.424	0.483
Gluconate 2-dehydrogenase (acceptor) inhibitor	0.578	0.714	0.916	0.733
Acute neurologic disorders treatment	0.502	0.558	0.809	0.669
Ubiquinol-cytochrome-c reductase inhibitor	0.666	0.522	0.575	0.579
Kidney function stimulant	0.453	0.444	0.62	0.443
Vasodilator, peripheral	0.474	0.587	0.46	0.398
Fibrinolytic	0.526	0.353	0.43	0.342
5-Hydroxytryptamine release stimulant	0.292	0.749	0.72	0.618
Kinase inhibitor	0.364	0.655	0.743	0.612

**Table 6 pharmaceuticals-17-01595-t006:** Indolylquinone toxicity risks and physicochemical properties.

Properties		Molecules			
		IA	IB	IC	ID
Toxicity Risks	Mutagenic	N	H	N	N
	Tumorigenic	N	N	N	N
	Irritant	N	N	N	N
	reproductive effect	N	N	N	N
Physicochemical Properties	clog *P*	2.93	1.24	0.38	1.68
	Solubility (log *S*)	−3.16	−2.6	−2.34	−2.72
	Mol weight	279.34	283.28	255.23	297.31
	TPSA	49.93	68.39	90.39	68.39
	Drug likeness	0.33	2.75	2.88	2.23
	H bond acceptor	3	5	5	5
	H bond donor	1	1	3	1
	Nb stereocenters	0	0	0	0
	Nb rotatable bonds	2	3	1	3
	Drug Score	0.73	0.56	0.94	0.9

N = no risk, H = High risk.

**Table 7 pharmaceuticals-17-01595-t007:** Energy values of the interaction (score) between the ligands with the protein.

Properties	Reference		IA	IB		IC			ID	
Binding Energy	−11.69		−8.75	−7.75		−7.29			−8.13	
Ligand Efficiency	−0.42		−0.42	−0.37		−0.38			−0.37	
Inhibition constant (μM)	2.69		378.72	2.08		4.55			1.11	
Intermolecular energy	−13.18		−9.34	−8.65		−8.18			−9.02	
vander Waals, hydrogen bond, and desolvation energies	−12.77		−9.28	−8.56		−7.92			−8.99	
Electrostatic energy	−0.41		−0.06	−0.09		−0.27			−0.03	
Total internal	−0.66		−0.91	−0.76		−1.86			−0.81	
Torsional energy	1.49		0.6	0.89		0.89			0.89	
Unbound energy	−0.66		−0.91	−0.76		−1.86			−0.81	
Hydrogen bond number	2		1	2		3			2	
Type	C=O−Gly202		C=O−Arg217	C=O−Tyr235		N−H−Gly202			CH3−O−Met229	
Distance (A°)	2.179	2.021	1.907	2.148	2.191	H−O−ILE234	H−O−Asp105	N−H−Asp109	N−H−Gly202	CH3−O−Met229
Angle D–H–A (°)	150.466	176.831	160.562	154.29	147.493	2.237	2.213	2.064	2.132	2.188
Energy	−3.083	−6.143	−6.026	−3.725	−2.633	123.509	133.158	153.495	160.985	147.752

**Table 8 pharmaceuticals-17-01595-t008:** Prediction of the absorption and distribution of the target molecules in different models.

	Model	Compounds	Results	Probability
Absorption	Human intestinal absorption (HIA)	IA	+	0.9974
		IB	+	0.9914
		IC	+	0.9933
		ID	+	0.9931
	Human Oral bioavailability (HOB)	IA	+	0.6571
		IB	+	0.7429
		IC	+	0.7286
		ID	+	0.8286
	Caco-2 permeability	IA	+	0.8133
		IB	+	0.7444
		IC	−	0.7145
		ID	+	0.7637
Distribution	P-glycoprotein substrate	IA	−	0.844
		IB	−	0.7849
		IC	−	0.8885
		ID	−	0.9073
	P-glycoprotein inhibitor	IA	−	0.7651
		IB	−	0.571
		IC	−	0.9407
		ID	−	0.6178
	Blood–brain barrier (BBB) penetration	IA	+	0.9745
		IB	+	0.9407
		IC	+	0.9227
		ID	+	0.9407

**Table 9 pharmaceuticals-17-01595-t009:** Metabolism and toxicity prediction of the studied indolylquinones.

	Models	Compound	Results	Probability
Metabolism	CYP2C9 Substrate	IA	−	0.592
		IB	−	1.000
		IC	−	0.809
		ID	−	1.000
	CYP2D6 Substrate	IA	−	0.869
		IB	−	0.824
		IC	−	0.862
		ID	−	0.853
	CYP3A4	IA	+	0.577
		IB	+	0.569
		IC	+	0.558
		ID	+	0.557
	CYP1A2	IA	+	0.893
		IB	+	0.821
		IC	+	0.754
		ID	+	0.810
	CYP2D6	IA	−	0.685
		IB	−	0.828
		IC	−	0.829
		ID	−	0.861
	CYP2C9 inhibitor	IA	+	0.841
		IB	+	0.777
		IC	+	0.537
		ID	+	0.792
Toxicity	Eye corrosion	IA	−	0.993
		IB	−	0.990
		IC	−	0.992
		ID	−	0.991
	Ame’s mutagenesis	IA	−	0.580
		IB	+	0.580
		IC	−	0.520
		ID	−	0.540
	Carcinogenesis	IA	−	0.932
		IB	−	0.820
		IC	−	0.886
		ID	−	0.932

## Data Availability

The original contributions presented in the study are included in the article, further inquiries can be directed to the corresponding authors.

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
