# Peer review of "Theoretical–Cheminformatic Study of Four Indolylphytoquinones, Prospective Anticancer Candidates"

_pharmaceuticals, 2024, doi:10.3390/ph17121595_

Round 1
Reviewer 1 Report
Comments and Suggestions for Authors
In the publication, the authors performed studies based on in silico modelling of four indolylquinones, of potential compounds as anticancer drugs. Physicochemical analysis of these compounds was performed to predict their pharmacological properties. The stages of the study were based on: optimization of the molecular structures of the compounds described as IA-ID (DFT) to evaluate the stability and reactivity of the proposed molecules. The analysis of physicochemical parameters was presented and demonstrated their compliance with Lipinski and Veber's rules, which may indicate that IA-ID compounds can show good absorption and membrane penetrability through biological membranes. Molecular docking was also performed, and PARP-1 protein was chosen as the target to evaluate potential interactions with this apoptosis-related biological target, suggesting a possible anticancer effect. After a series of studies, the authors concluded that only IC and IA compounds, show good reactivity and may have potential anticancer activity, as suggested by the predicted IC50 values (about 25.0 mg/mL).
The present work is an interesting and developmental continuation of the authors' previous research related to synthesis (https://doi.org/10.1155/2016/3870529) and biological and in silico modeling-based studies (https://doi.org/10.3390/molecules22071060) of compounds based on the indolylquinone backbone.
I regard in silico research as extremely valuable because it enables us to approximate the physicochemical properties, as well as molecular interactions, of the designed compounds, which can further enhance the process of designing new drugs.
Nevertheless, I believe that certain points in the work should be better explained/presented.
In the Introduction section, the authors should better outline their goal for the reader. Why did they choose model compounds like TQ, DMQ, DHQ, DMMQ for further modifications? Quinones, considered to be among the most potent pharmacophores their molecular mechanisms of cytotoxicity have been extensively studied, I find this lacking in the text and literature attachments. In addition, what are the biological activities, metabolic and physicochemical properties of indolylquinones known from the literature?
Please pay attention to this part of the text, where you can get confused with the authors' designation of compounds:
Line 84-85 can be written differently: the targets evaluated are derivatives thymoquinone (TQ), 2,6-dimethoxy-1,4- benzoquinone ( DMQ), 2,3-dimethoxy-5-methyl-p-benzoquinone (DHQ), and 2,5-dihydroxy-1,4-benzoquinone (DMMQ) referred to as IA-ID in Scheme 1, respectively.
Please correct Scheme 1 to make it more readable, separate more clearly the group of benzoquinone compounds from IA-ID compounds. IB does not have a signature in this scheme.
Many quinones exhibit poor bioavailability and low solubility in aqueous systems, which hinders their in vivo systemic application and reduces their therapeutic value. In the Results and discussion section of the paper, I am missing a comparison of the starting properties of quinones with the tested compounds (IC50, lipophilicity, bioavailability, etc.). There is no commentary on whether the authors' proposed modifications were accurate or not.
There are numerous editorial errors in the text: inadequate font, improper use of lowercase/uppercase letters, and incorrect use of punctuation marks. I note that these items should be corrected to improve readability and compliance with the editorial requirements of the journal. (e.g., line 122, 185, table 1, line 298, table 6, table 5).
Author Response
Reviewer 1
- Reviewer: In the Introduction section, the authors should better outline their goal for the reader. Why did they choose model compounds like TQ, DMQ, DHQ, DMMQ for further modifications?
Us: Done, please review the modified manuscript
- Reviewer: Quinones, considered to be among the most potent pharmacophores their molecular mechanisms of cytotoxicity have been extensively studied, I find this lacking in the text and literature attachments. In addition, what are the biological activities, metabolic and physicochemical properties of indolylquinones known from the literature?
Us: Done, please review the modified manuscript
Please pay attention to this part of the text, where you can get confused with the authors' designation of compounds:
- Reviewer: Line 84-85 can be written differently: the targets evaluated are derivatives thymoquinone (TQ), 2,6-dimethoxy-1,4- benzoquinone ( DMQ), 2,3-dimethoxy-5-methyl-p-benzoquinone (DHQ), and 2,5-dihydroxy-1,4-benzoquinone (DMMQ) referred to as IA-ID in Scheme 1, respectively. Please correct Scheme 1 to make it more readable, separate more clearly the group of benzoquinone compounds from IA-ID compounds. IB does not have a signature in this scheme.
Us: Done, we correct figure 1.
- Reviewer: Many quinones exhibit poor bioavailability and low solubility in aqueous systems, which hinders their in vivo systemic application and reduces their therapeutic value. In the Results and discussion section of the paper, I am missing a comparison of the starting properties of quinones with the tested compounds (IC50, lipophilicity, bioavailability, etc.). There is no commentary on whether the authors' proposed modifications were accurate or not.
Us: Done, We have attended to the request
- Reviewer: There are numerous editorial errors in the text: inadequate font, improper use of lowercase/uppercase letters, and incorrect use of punctuation marks. I note that these items should be corrected to improve readability and compliance with the editorial requirements of the journal. (e.g., line 122, 185, table 1, line 298, table 6, table 5).
Us: Done, we correct the errors.

Reviewer 2 Report
Comments and Suggestions for Authors
Despite the significant successes of pharmacologists in the development of various drugs, the pharmaceutical industry is still constantly searching for new substances that differ from the known ones in lower toxicity, greater selectivity, targeted effect on biological targets and many other qualities, including overcoming the effects of multiple drug resistance. From this point of view, the article presented by the authors on the study of four 3-indolylquinones with possible antitumor activity is relevant and practically significant. The authors used a modern methodological apparatus for in silico studies of new substances, and here, too, there are no special complaints, meanwhile, there are significant obstacles to giving a positive assessment of this work as a whole.
1. There is no clearly substantiated idea of ​​choosing these four 3-indolylquinones from a hypothetical series containing tens of thousands of substances containing this motif. Why these four? 2. These compounds were not even synthesized for simple lipophilicity studies, not to mention more specialized extended tests for biological activity in vitro and in vivo.
Unfortunately, all of the above does not allow me to respond positively to the approval of this article in the authoritative journal Pharmaceuticals, perhaps the authors should send this material to a more specialized journal.
Author Response
Reviewer 2
- Reviewer: There is no clearly substantiated idea of ​​choosing these four 3-indolylquinones from a hypothetical series containing tens of thousands of substances containing this motif. Why these four?
Us: Done, We have improved the explanation of why quinones are evaluated
- Reviewer: These compounds were not even synthesized for simple lipophilicity studies, not to mention more specialized extended tests for biological activity in vitro and in vivo.
Us: The importance of this type of theoretical studies is explained in the manuscript. Please review the manuscript.

Reviewer 3 Report
Comments and Suggestions for Authors
Moyers-Montoya and colleagues have prepared a manuscript on a theoretical-cheminformatic study of four indolylphyto-quinones, prospective anticancer candidates. As follows from the abstract of the article, the manuscript is devoted to an in silico study of four new hypothetical 3-indolylquinones with possible antitumor activity in comparison with indolylperezone, indolylisoperezone, indolylplumbagin, and indolylmenadione. Overall, the manuscript is clear; however, in my opinion, it does not correspond to the topic of the special issue to which it is submitted. In addition, the manuscript is poorly structured and the main questions it addresses are not clearly stated and will be difficult for the readership to understand in its current form.
Abstract – does not give a clear idea of ​​the results obtained by the authors. In addition, the last 2 sentences need to be formulated differently.
The introduction consists of chaotic and unfounded insertions of various information from literary sources, which in no way help to understand the relevance of the study. It is necessary to clearly formulate the originality of the work. Why did the authors choose these molecules? Why are indolylquinones important, what role does this residue play in the manifestation of activity? What is the activity of the original quinones shown in Scheme 1? How important is the approach used by the authors (in silico) to obtain information related to their biological, metabolic activity and physicochemical properties?
Scheme 1 – should be called Figure 1, because the figure does not show any reactions. In addition, the title contains errors and the numbering of the compounds is not indicated everywhere
Line 50-52 – a literary source?
Line 53-55 is an unclear sentence
Results and discussion – it is not entirely clear why the authors keep returning to their previous study (literature reference 11). If for comparison, then this should be formulated more clearly, especially since a whole section and table 3 are devoted to the previously described compounds.
Line 203-204 – what literature data supports this statement?
Materials and Methods – all analyses presented are properly formatted
Conclusions – do not correspond to the presented evidence and arguments. The authors state that the purpose of their study is to provide recommendations based on the results of a theoretical-computational chemoinformatic contribution of four indolylquinones, accomplishing information related to their biological, metabolic activity and physicochemical properties; however, in seven lines of conclusions, there is no summary of these recommendations. In addition, the conclusions should contain the authors' conclusions on the future prospects of the results obtained, which is also missing.
Author Response
Reviewer 3
- a) Reviewer: Abstract – does not give a clear idea of ​​the results obtained by the authors. In addition, the last 2 sentences need to be formulated differently.
Us: Done, the abstract was improved, please review the manuscript.
- b) Reviewer: The introduction consists of chaotic and unfounded insertions of various information from literary sources, which in no way help to understand the relevance of the study. It is necessary to clearly formulate the originality of the work. Why did the authors choose these molecules? Why are indolylquinones important, what role does this residue play in the manifestation of activity? What is the activity of the original quinones shown in Scheme 1? How important is the approach used by the authors (in silico) to obtain information related to their biological, metabolic activity and physicochemical properties?
Us: Done, the introduction and outline have been improved, please review the manuscript. Furthermore, further explanation is given of the importance of in silico studies.
- c) Reviewer: Scheme 1 – should be called Figure 1, because the figure does not show any reactions. In
addition, the title contains errors and the numbering of the compounds is not indicated everywhere
Us: Done, the word scheme was changed to figure.
- Reviewer: Line 50-52 – a literary source?
Us: Done, the reference was added
- Reviewer: Line 53-55 is an unclear sentence
Us: done, the sentence has been rewritten
- Reviewer: Results and discussion – it is not entirely clear why the authors keep returning to their previous study (literature reference 11). If for comparison, then this should be formulated more clearly, especially since a whole section and table 3 are devoted to the previously described compounds.
Us: The results section was modified to indicate comparisons with similar compounds.
- Reviewer: Line 203-204 – what literature data supports this statement?
Us: Done, the corresponding reference was added
- Reviewer: Conclusions – do not correspond to the presented evidence and arguments. The authors state that the purpose of their study is to provide recommendations based on the results of a theoretical-computational chemoinformatic contribution of four indolylquinones, accomplishing information related to their biological, metabolic activity and physicochemical properties; however, in seven lines of conclusions, there is no summary of these recommendations. In addition, the conclusions should contain the authors' conclusions on the future prospects of the results obtained, which is also missing.
Us: done, the conclusions were modified

Reviewer 4 Report
Comments and Suggestions for Authors
In the manuscript entitled “Theoretical-cheminformatic study of four indolylphyto-quinones, prospective anticancer candidates,” the authors considered four hypothetical compounds, whose structure represents a benzoquinone core to which an indole segment is attached. It is known that naturally occurring quinones have significant biological activity, which, judging by literature data, increases due to conjugation with an indole ring. Therefore, the idea of ​​the manuscript is to use a theoretical approach to evaluate the pharmacokinetic potential of these four hypothetical molecules in order to predict the importance of their potential synthesis, with the aim of using them in medical treatments. For this purpose, the authors used the following methodological approaches: Optimization of the molecules structure is done by using the DFT method, as well as calculation of the reactivity descriptors and MEP analysis; the biological activities are predicted by using the chemoinformatic PASS server, while toxicology and physicochemical properties are determinated by using the OSIRIS explorer; the inhibition capacity of molecules toward PARP-1 protein (PDB ID: 1UKO) is investigated using the molecular docking approach; pharmacological and metabolic properties are predicted by using ADMET analysis.
The conceptualization of the investigations as well as the methodology approach is very correct. The writing style is very clear and concise. The results are interesting, and presented correctly. The literature data are correctly used and cited.
With everything mentioned in mind, I suggest publishing this review article after following changes, mostly of a technical nature:
1. The Results and discussion section should be separated into parts DFT calculation, Molecular docking and Pharmacokinetic and toxicological properties, due to easier results following.
2. There is no written which computer program is used for DFT calculations. The applied program should be highlighted and cited correctly.
3. What label letters a, b, c and d in the superscript in Table 1?
4. Tre are a lot of typo errors:
· Line 45: There is no blank space between 30 and the parentheses, 30[1,2].
· Line 52: Two In (… In In this sense…). After parentheses should be full stop.
· Line 64: Excess empty space in the: to recommend
· The title of Scheme 1, line 91, after a comma should be empty space (TQ,DMQ, …)
· Line 122: There misses parentheses ([21;…)
· Line 189: It is written IE-II. What is II? I believe it should be IH.
· Line 298: The title of Table 6 should start with the capital letter I.
· Lines 346, 365, 369, 373: There is somewhere the excess empty space, and somewhere it misses.
· Table 9, Toxicity, Eye corrosion for molecule IB, there is written 9898, it should be 0.9898.
· Tables 1 and 6: The used font should be uniformed with the font used in the rest of the text.
Author Response
Reviewer 4
- Reviewer: There is no written which computer program is used for DFT calculations. The applied program should be highlighted and cited correctly.
Us: The implemented program was appropriately highlighted and cited.
- Reviewer: What label letters a, b, c and d in the superscript in Table 1?
Us: Finished, the label letters a, b, c and d were explained.
- Reviewer: Tre are a lot of typo errors:
- Line 45: There is no blank space between 30 and the parentheses, 30[1,2].
- Line 52: Two In (… In In this sense…). After parentheses should be full stop.
- Line 64: Excess empty space in the: to recommend
- The title of Scheme 1, line 91, after a comma should be empty space (TQ,DMQ, …)
- Line 122: There misses parentheses ([21;…)
- Line 189: It is written IE-II. What is II? I believe it should be IH.
- Line 298: The title of Table 6 should start with the capital letter I.
- Lines 346, 365, 369, 373: There is somewhere the excess empty space, and somewhere it misses.
- Table 9, Toxicity, Eye corrosion for molecule IB, there is written 9898, it should be 0.9898.
- Tables 1 and 6: The used font should be uniformed with the font used in the rest of the text.
Us: Done, the comments were addressed

Round 2
Reviewer 2 Report
Comments and Suggestions for Authors
I am satisfied with the authors' response.
Author Response
Reviewer: I am satisfied with the authors' response.
Us: Thank you for your comments

Reviewer 3 Report
Comments and Suggestions for Authors
The authors have considered almost all the reviewer's comments and made the appropriate corrections to the text of the article. However, I still have some comments - lines 43,44 and 557,558 - the final sentences in the abstract and conclusions are unclear, do not make sense and should either be deleted or rewritten appropriately. The conclusions should contain the authors' justification for future prospects and the possibility of practical application of the results obtained.
Author Response
Reviewer 2: The authors have considered almost all the reviewer's comments and made the appropriate corrections to the text of the article. However, I still have some comments - lines 43,44 and 557,558 - the final sentences in the abstract and conclusions are unclear, do not make sense and should either be deleted or rewritten appropriately. The conclusions should contain the authors' justification for future prospects and the possibility of practical application of the results obtained.
Us: Done, the abstract and conclusión was improved, please review the manuscript.
